# Geometric isotope effect of deuteration in a hydrogen-bonded host–guest crystal

Chao Shi[1], Xi Zhang[2], Chun-Hua Yu[1], Ye-Feng Yao [2,3] & Wen Zhang [1]

Deuteration of a hydrogen bond by replacing protium (H) with deuterium (D) can cause geometric changes in the hydrogen bond, known as the geometric H/D isotope effect (GIE). Understanding the GIEs on global structures and bulk properties is of great importance to study structure–property relationships of hydrogen-bonded systems. Here, we report a hydrogen-bonded host–guest crystal, imidazolium hydrogen terephthalate, that exemplifies striking GIEs on its hydrogen bonds, phases, and bulk dielectric transition property. Upon deuteration, the donor–acceptor distance in the O–H···O hydrogen bonds in the host struc-ture is found to increase, which results in a change in the global hydrogen-bonded supra-molecular structure and the emergence of a new phase (i.e., isotopic polymorphism). Consequently, the dynamics of the confined guest, which depend on the internal pressure exerted by the host framework, are substantially altered, showing a downward shift of the dielectric switching temperature.

[1] Ordered Matter Science Research Center and Jiangsu Key Laboratory for Science and Applications of Molecular Ferroelectrics, Southeast University, 211189 Nanjing, China. [2] Department of Physics & Shanghai Key Laboratory of Magnetic Resonance, School of Physics and Materials Science, East China Normal University, North Zhongshan Road 3663, 200062 Shanghai, China. [3] NYU-ECNU Institute of Physics at NYU Shanghai, 3663 Zhongshan Road North, 200062 Shanghai, China. Correspondence and requests for materials should be addressed to Y.-F.Y. (email: yfyao@phy.ecnu.edu.cn) or to W.Z. (email: zhangwen@seu.edu.cn)

Deuteration is a chemical tool to provide minimal modification of molecules by replacing a protium atom (H) with a deuterium atom (D). Due to their differences in atomic mass, volume, and spin, the deuteration can exert significant influences on the chemical and physical properties of molecules, such as molecular spectroscopic signatures, kinetics, and equilibrium constants[1–3]. However, beyond the scope of molecules, the H/D isotope effect on molecular aggregates (i.e., crystal structures) and related bulk properties is not straightforward.

A long-held assumption is that deuteration generally does not alter crystal structures[4]. This is true in most cases; however, there emerge several exceptions, which unjustifies this assumption. These exceptions have shown that different crystal structures, including oxalic acid dihydrate, a complex of pentachlorophenol and 4-methylpyridine, pyridine, and other[5–7], upon deuteration, exhibit a phenomenon known as "isotopic polymorphism"[5]. The occurrence of isotopic polymorphism in crystals seems serendipitous and unpredictable[8]. In particular, the relationship and transformation among the polymorphs under external stimuli (e.g., temperature and pressure) have been less investigated; however, these investigations could provide key clues to understand the interplays among intermolecular interactions in crystal packing (a central issue in crystal engineering). Hydrogen (H) bonds, as supramolecular synthons[9], play a vital role in crystal engineering and become a more predictable approach to exhibit isotopic polymorphism. This is because deuteration of the H bond can cause geometric changes in the bond, which is known as the geometric H/D isotope effect (GIE)[10, 11]. This effect accumulates and transmits in the bulk structures of phases.

The influence of the H/D isotope effect on the phase-related bulk properties also remains uncertain. A critical focus is structural phase transitions, which are often associated with the occurrence of bulk properties, such as ferroelectricity/antiferroelectricity[12–16] and dielectric transitions[17]. In molecular dielectrics, manipulation and control of the dynamic order–disorder transitions of the components (molecules or ions), which are dominated by hierarchical intermolecular interactions, are very challenging tasks for the design and modification of these materials and have been long pursued. In some hydrogen-bonded ferroelectrics[14, 18, 19], deuteration of H bonds can cause significant shifts of the phase transition temperatures ($T_{tr}$). For example, $KD_2PO_4$ shows a much higher $T_{tr}$ than $KH_2PO_4$, in which the huge isotope effect (a temperature change of 107 K) has been associated with proton tunneling and/or a geometrical effect[20–23]. Furthermore, some other bulk physical properties, such as thermal behavior, elasticity, electroluminescence, magnetism, and even superconducting transition temperature[24–27], can also be significantly influenced upon deuteration. In contrast, there are also many examples showing negligible isotope effects, such as the $T_{tr}$ in some H-bonded ferroelectrics[28] and switchable dielectrics[29]. These seemingly opposite phenomena suggest complex relationships between the H/D isotope effect and the bulk properties.

From a structural viewpoint, the bulk properties rely on a specific molecular arrangement or crystal packing rather than the molecule itself. Therefore, the mechanism for deuteration of the molecules altering the intermolecular interactions and the phases is key to understanding and evaluating the H/D isotope effect on these bulk properties. However, comprehensive investigations on molecular structure–phase–property relationships of H/D isotope effects are very rare due to a lack of suitable model systems[18, 19, 30]. Our work focuses on molecule-based dielectrics that undergo structural phase-transition-driven dielectric transitions between relatively low-dielectric and high-dielectric states (LDS and HDS)[17, 31–33], a type of recently identified switchable

dielectrics as the electrical counterpart to magnetic spin crossover materials[34, 35]. The $T_{tr}$, as a key parameter to describe the dielectric switching temperature, is determined by motional changes of polar molecules between orientationally ordered (frozen) and dynamically disordered (melt-like) states in a crystal lattice[17, 32].

Herein, we establish a H-bonded host–guest model system, (Im)(TPA) (1; Im = imidazolium cation, TPA = hydrogen terephthalate monoanion), to demonstrate significant GIEs on H bonds, the phases (two and three phases before and after deuteration, respectively), and the bulk dielectric transition properties under variable-temperature conditions. The basic structural unit to exhibit the GIE is the characteristic short COO–H⋯OOC H bonds in the anionic channel-like host composed of TPA. Upon deuteration, significant geometric changes of the H bonds and the host structure modulate the dynamics and arrangement of the Im guest in the host channel, leading to isotopic polymorphism and a downward shift of the $T_{tr}$. The driving force for these changes is the GIE-induced internal (chemical) pressure on the host framework, which transmits this effect to the confined guest via intermolecular interactions.

## Results

**Synthesis and phase transitions.** Crystals of **1** were grown from a solution containing imidazole and terephthalic acid. To study the GIE, we prepared the fully protonated compound **1-d₀** and three partially deuterated isotopologs **1-d₂**, **1-d₃**, and **1-d₅** (Fig. 1). According to the deuteration of the COO–H⋯OOC dimeric H bonds, the four compounds are classified as H isomorphs (**1-d₀** and **1-d₃**) or D isomorphs (**1-d₂** and **1-d₅**). They are stable up to 430 K, as shown by thermogravimetric analysis (TGA) (Supplementary Fig. 1).

The phase transitions and H/D isotope effects in **1** were first revealed by differential scanning calorimetry (DSC) measurements (Fig. 2a and Supplementary Fig. 2a). The H isomorphs

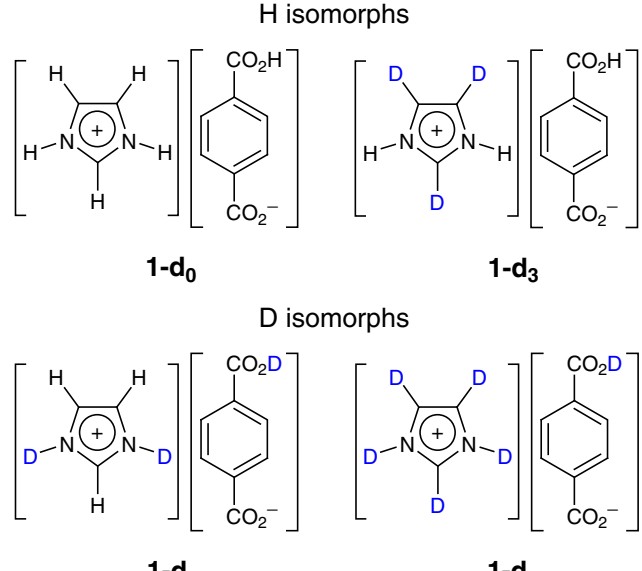

**Fig. 1** Chemical structure of **1**. Isotopologs **1-d₀**, **1-d₂**, **1-d₃**, and **1-d₅** (the subscript referring to the number of deuterium atoms in Im) are classified into H isomorphs and D isomorphs according to the deuteration state of the carboxyl group COOH. Note: Isotopologs refer to a group of compounds differing in the number and/or position of atom substitutions by isotopes, while isomorphs refer to crystals belonging to the same space group

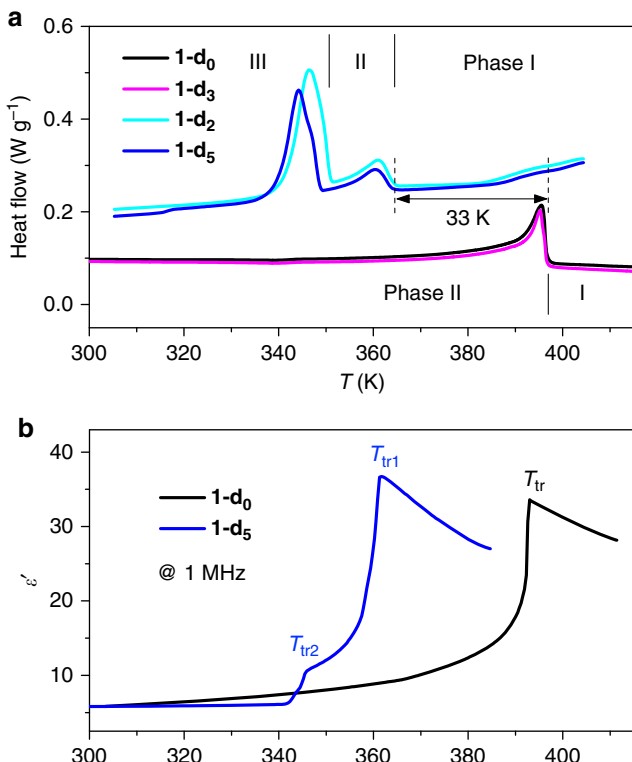

**Fig. 2** Phase and dielectric transitions. **a** Differential scanning calorimetry curves of **1** upon cooling, showing two phases for the H isomorphs (**1-d$_0$** and **1-d$_3$**) and three phases for the D isomorphs (**1-d$_2$** and **1-d$_5$**). **b** The real parts of the dielectric constants $\varepsilon'$ of **1-d$_0$** and **1-d$_5$** measured on crystal plates upon cooling, showing dielectric switching between low-dielectric and high-dielectric states. The largest dielectric responses appear on the [010] crystal plane, which is a cleavage plane and designated according to the crystal structures at 413 K in phase I

showed the same thermal behavior with a $T_{tr}$ centered at 397 K. The phases above and below the $T_{tr}$ are designated phase I and II, respectively. The D isomorphs showed strikingly different thermal behavior from the H isomorphs. A new phase (phase III below $T_{tr2} = 350$ K) emerges besides phase I and II (with $T_{tr1}$), corresponding to isotopic polymorphism. Phase II remains stable within a narrow temperature window of approximately 14 K. The downward shift in $T_{tr1}$ is approximately 33 K for the I–II phase transition from the H to D isomorphs. The I–II and II–III phase transitions in **1-d$_5$** show second-order and first-order characteristics, respectively (Supplementary Fig. 2b, c). Comparing the $T_{tr}$ values of the four isotopologs reveals that deuteration of the CH group of the Im cation does not affect the phases and $T_{tr}$, in contrast to deuteration of the OH and NH groups involved in H-bonding.

**GIE on the dielectric switching property.** The dielectric constant ($\varepsilon = \varepsilon' - i\varepsilon''$) of a molecular material microscopically relates mainly to the orientational polarization of the polar components in the low-frequency range (within MHz)[36]. Thermal responses of the dielectric constant not only provide insight into the mechanism of the dynamics of dipoles but also lead to a type of responsive dielectric materials with dielectric switching between relatively LDS and HDS[17].

The distinct dielectric transitions of **1** were compared between crystal (all discussed crystal directions were based on the structures at 413 K in phase I for ease of comparison) and

powder samples (Fig. 2b and Supplementary Fig. 3). The dielectric transition temperatures coincided with the DSC results (Fig. 2a). The real part $\varepsilon'$ of **1-d$_0$** measured along the [010] cleavage plane shows a dielectric transition at the $T_{tr}$ with a change from approximately 6 at 300 K to a peak value of 34, corresponding to a switching ratio of approximately 6. We ascribe the gradual rise in $\varepsilon'$ below the $T_{tr}$ to a pretransition effect[32]. Above the $T_{tr}$, $\varepsilon'$ begins to decrease, reflecting competition between the dipole alignment and thermal disturbance. The normalized $\varepsilon'$ value shows dielectric switching around the $T_{tr}$ between the LDS (off) and HDS (on) (Supplementary Fig. 4). In contrast, for the as-grown (001) plate, $\varepsilon'$ remains in a low state with a gradual increase from 6.8 at 300 K to 8.3 above the $T_{tr}$. This striking dielectric anisotropy indicates that the local polarization changes are confined in the [010] or *b* direction, which can be attributed to reorientations of the Im cation, as discussed below. The other H-isomorph **1-d$_3$** shows the same behavior as that of **1-d$_0$** (Supplementary Fig. 3).

The D isomorphs **1-d$_2$** and **1-d$_5$** exhibit different dielectric transitions from those of the H isomorphs (Fig. 2b and Supplementary Fig. 3). A small change in $\varepsilon'$ (from 6 at 340 K to 11 at 345 K) of approximately 5 occurs at $T_{tr2}$, corresponding to the II–III transition, which is thought to be due to a lattice distortion and relative displacement of the molecules. Then, $\varepsilon'$ quickly increases to its maximum value of approximately 37 at $T_{tr1}$, corresponding to the I–II transition. The latter dielectric transition is similar to those of the H isomorphs, indicating that they share the same microscopic structural origin. However, the downward shift of approximately 33 K in $T_{tr1}$ reflects the H/D isotope effect on the dynamics of the guest, indicating a different mechanism from that of the H-bond-triggered properties in most reported ferroelectrics[14, 18, 19].

**Crystal structures and isotopic polymorphism.** To clarify the structural origins of the phase transitions and GIE, variable-temperature single-crystal X-ray diffraction analysis was performed on the phases of **1** (Supplementary Table 1). The H isomorphs, taking **1-d$_0$** as an example, were found to crystallize in space group *Pnma* in phase I and $P2_1/c$ in phase II. At 293 K (phase II), the carboxylate group (O1–C–O2) of TPA is in the plane of the benzene ring, whereas the carboxyl group (O3–C–O4) is obviously twisted from the ring plane with a dihedral angle of 15° (Fig. 3a). The hexagonal-like packing of the end-to-end-linked TPA monoanions is stabilized by H bonds ($d_{O1\cdots O3}$ of 2.469 Å) and $\pi\cdots\pi$ interactions (centroid distance of 3.845 Å) between neighboring TPA anions (Fig. 3b; Supplementary Tables 2–3), forming supramolecular channels (Fig. 3c, d and Supplementary Fig. 5). The donor–acceptor distance $d_{O\cdots O}$ is characteristic of short/strong H bonds ($d_{O\cdots O}$ of 2.4–2.5 Å)[37–40]. The Im cations residing in the channel adopt two orientations with a dihedral angle of approximately 89° and are approximately perpendicular to the TPA ring plane with a dihedral angle of approximately 81°. Weak intermolecular interactions between the guest and host can be seen in the **1-d$_0$** structures derived by the Hirshfeld surface analysis[41, 42] (Fig. 3c and Supplementary Fig. 6). In phase I (413 K; Supplementary Fig. 5), all atoms in TPA are co-planar; however, the O4 atom shows an elongated thermal ellipsoid perpendicular to the ring plane, which is evidence of thermal oscillations along the *b*-axis. The anionic chain remains stable with slightly increased intermolecular $d_{O\cdots O}$ (2.492 Å) and centroid separation (3.930 Å) values. More strikingly, the Im ring is bisected by mirror symmetry perpendicular to the *b*-axis and refined by a two-site disordered model with two orientations.

In addition to phases I and II, the D isomorphs show the additional phase III ($P2_1/n$) below 350 K, displaying a rare case of

isotopic polymorphism in the measured temperature range of 113–350 K. For **1-d₅**, the new phase has a doubled unit cell compared to that of phase II (Fig. 4 and Supplementary Figs. 7–8; Supplementary Table 1). At 293 K, the two carboxylate groups of

one of the two TPA anions in the asymmetric unit are twisted from the ring plane with dihedral angles of 24° (which is strikingly large) and 9°. There are two crystallographically independent Im cations in the channels, both of which show

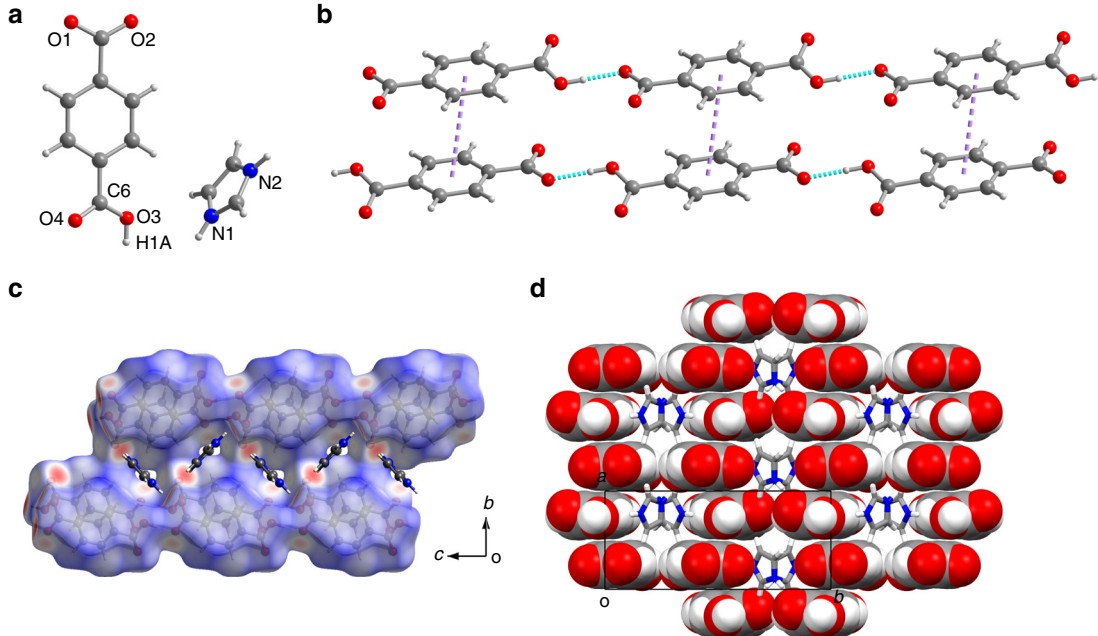

**Fig. 3** Crystal structure of **1-d₀** at 293 K. **a** Asymmetric unit. **b** Intermolecular interactions among TPA monoanions. Purple and cyan dotted lines represent the π⋯π and H-bonding interactions, respectively. **c** Cross-sectional view of the Hirshfeld surface of the anionic host channel containing the Im cations. The red, white, and blue regions of the surfaces correspond to the positive (close contact), neutral, and negative isoenergies, respectively. **d** Front view of the host–guest supramolecular channel structure

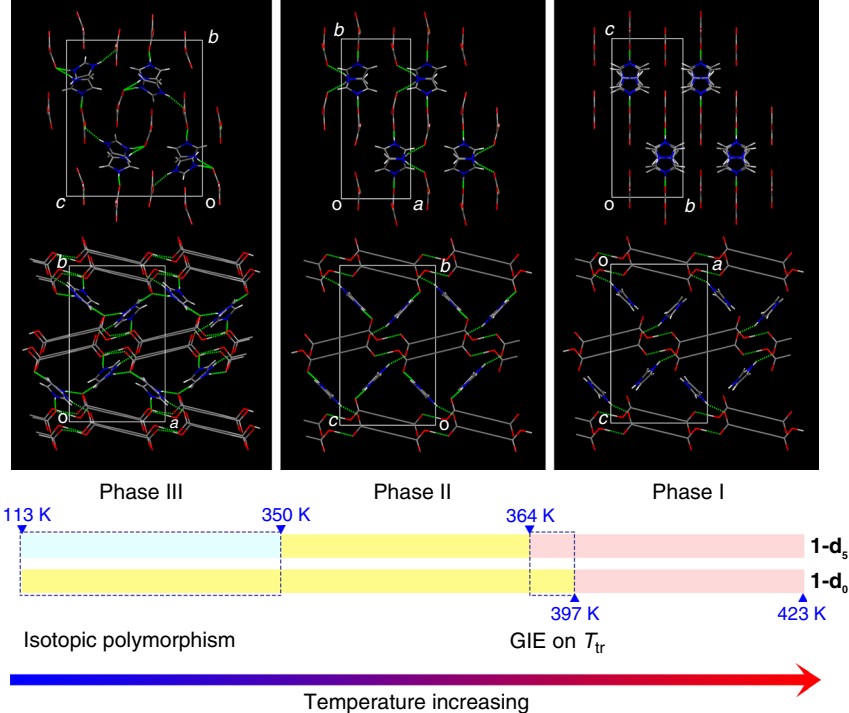

**Fig. 4** Structures of the different phases. Variation in the H-bonding networks in the three phases (phase I, light pink; phase II, yellow; phase III, light cyan), shown in a front view (upper row) and cross-sectional view (lower row) to the host channel. **1-d₀** and **1-d₅** are selected to represent the H and D isomorphs, respectively. The benzene ring in TPA is schematically drawn as a long C–C single bond for clarity. Red: O; blue: N; gray: C; light gray: H/D. The H bonds are represented as dotted lines. Isotopic polymorphism occurs below 350 K (phase III). The difference in the I–II phase transitions between **1-d₀** and **1-d₅** reveals the H/D isotope effect on $T_{tr}$ (a downward shift of approximately 33 K). Note: the temperature ranges are schematically illustrated

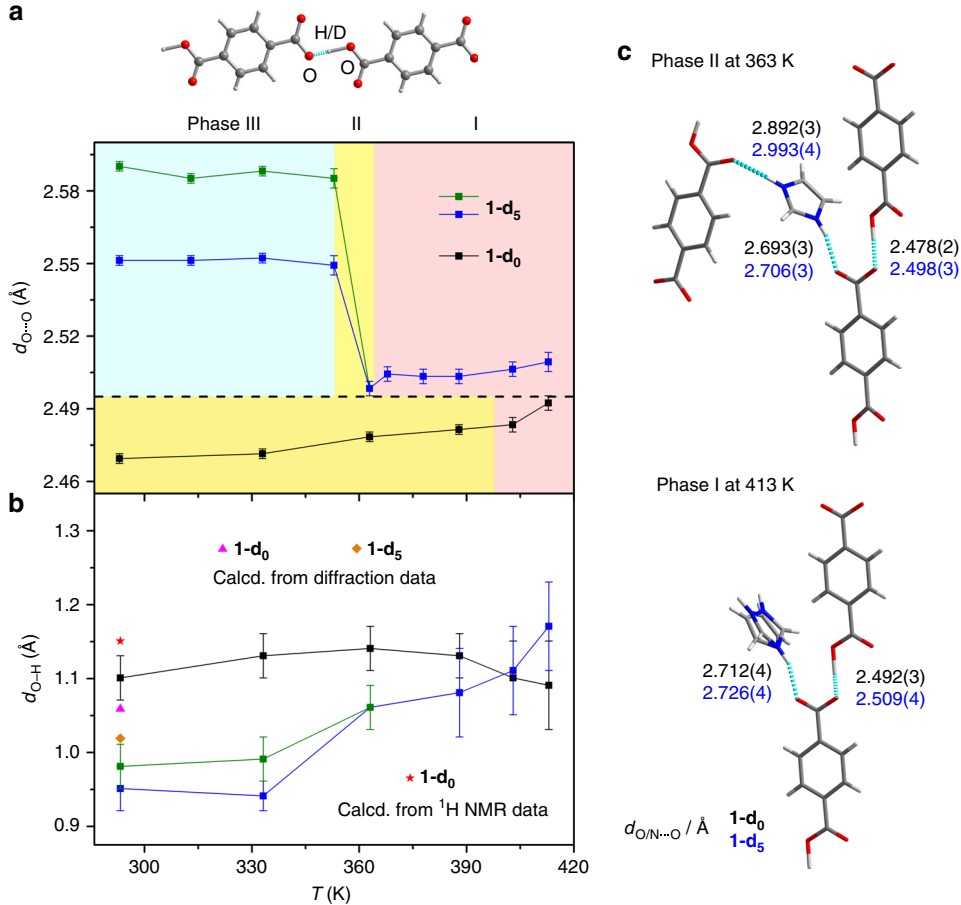

**Fig. 5** GIEs on the O–H···O H bonds between two TPA anions in different phases of **1-d$_0$** and **1-d$_5$**. **a** Secondary GIE on the O···O distance ($d_{O···O}$) (phase I, light pink; phase II, yellow; phase III, light cyan). Lines are placed to guide the eye. **b** Primary GIE on the O–H bond length ($d_{O-H}$). The measured diffraction data are compared with the calculated values from either the diffraction data or the [1]H NMR data at 293 K, based on the equations and parameters published by Limbach et al[43]. The error bars in **a** and **b** are the estimated standard deviations (esds) calculated rigorously from the full covariance matrix. The esds in the bond lengths output by SHELXL take the errors in the unit-cell dimensions into account. **c** Local H-bonded configurations of **1-d$_0$** and **1-d$_5$** in phase I (413 K) and II (363 K). Only the $d_{O/N···O}$ values are shown (black for **1-d$_0$** and blue for **1-d$_5$**)

marked positional and orientational changes compared to those in phase II. One of the Im cations is almost perpendicular to the TPA ring planes with dihedral angles of 90° and 85°, whereas the other Im shows dihedral angles of 67° and 62° to the TPA ring planes. The dihedral angle of the two adjacent Im cations is 82°, and their centroid–centroid distance is approximately 5.24 Å, which is much longer than the value of 4.99 Å for H-isomorph **1-d$_0$** in phase II at 293 K. The corresponding H-bond pattern of the Im cations with TPA anions is thus completely different from that in **1-d$_0$** (Fig. 4; Supplementary Table 2). In the H isomorph (**1-d$_0$**), each Im is H bonded with two TPA anions in adjacent chains through two N–H···O H bonds, with $d_{N···O}$ values of 2.675 (2) and 2.847(2) Å. However, in the D isomorph (**1-d$_5$**), the H-bond acceptors are changed to different TPA anions, and one of the O atoms is involved in two N–D···O bonds.

**GIEs on H bonds and supramolecular structures**. There are two types of GIEs in H bonds[11] (herein O–H···O): primary, which refers to a change in the position of H ($d_{O-H}$), and secondary, which refers to a change in the distance between the donor and acceptor ($d_{O···O}$) and which is also known as the Ubbelohde effect[10]. We first investigated the secondary GIE (the change in $d_{O···O}$ upon deuteration) from the X-ray diffraction data. For conciseness, a detailed comparison is provided only for **1-d$_0$** and **1-d$_5$** (Fig. 5a; Supplementary Table 2). In **1-d$_0$**, the $d_{O···O}$ value

gradually increases as the temperature is increased, from approximately 2.470 Å in phase II to approximately 2.483 Å in phase I, due to thermal expansion. There are no sharp changes during the I–II phase transition. In contrast, in **1-d$_5$**, $d_{O···O}$ shows an average elongation of 0.03 Å in phase I compared with **1-d$_0$**. This elongation is not large but is responsible for the alteration of the H bonds of the Im guest with the host (Fig. 5c; Supplementary Table 2), showing a weakening in the local H-bonding interactions (i.e., internal pressure) of the Im guest in **1-d$_5$** compared with those in **1-d$_0$**. This strikingly alters the dynamics of the Im guest and related dielectric switching properties of the D isomorphs, as shown above (a 33 K downward shift of the $T_{tr1}$). Furthermore, the II–III phase transition of **1-d$_5$** shows an increase in $d_{O···O}$ to approximately 2.550/2.587 Å (two types of $d_{O···O}$ emerge in phase III of the D isomorphs), with an increase of approximately 0.100 Å compared with that of **1-d$_0$**, resulting in the occurrence of isotopic polymorphism.

The primary GIE, that is, the position of the H/D atom in the O–H···O bond, is another key issue to fully understand the GIE on the H-bond geometry. In the difference Fourier maps of the diffraction data, one lobe of the electron density is found to be close to one of the two O atoms and can be assigned to the H/D atom (Supplementary Fig. 9). The distances between H/D and the two O atoms vary between approximately 1.1 and 1.4 Å (Fig. 5b; Supplementary Table 2) at different temperatures, indicating a

non-central location of the H/D atom in the O···O H bond. The significantly elongated O–H distance is commonly found in strong H bonds[37–40]. The primary GIE is evidenced by the bond length difference $\Delta d = d_{O-H} - d_{O-D}$, which yields values of +0.10 Å at 293 K and +0.08 Å at 363 K (that is, the O–H···O bond is less asymmetric than the O–D···O bond).

Using the correlation between $d_{O···O}$ and $(d_{O-H} - d_{H···O})/2$ without correction for the anharmonic zero-point motions[43–45] as an alternative method, the calculated $d_{O-H}$ is 1.058 Å for $d_{O···O}$ = 2.470 Å and $d_{O-D}$ is 1.018 Å for $d_{O···O}$ = 2.587 Å at 293 K, affording a roughly consistent but smaller value of $\Delta d$ = +0.04 Å than the corresponding diffraction result (Fig. 5b). Notably, the X-ray diffraction data locates the electron density maxima of the H/D atoms. Accurate positions of the nuclei in the H/D atoms can be obtained only by neutron diffraction studies. The two techniques are complementary, and future studies using neutron diffraction would afford a chance to compare the two results[46]. As a comparison, the primary GIE is also examined by correlating the solid-state [1]H NMR chemical shifts (Supplementary Fig. 10). The spectrum of 1-d₀ acquired at room temperature shows a clear peak for the O–H···O H bond at 19.8 ppm, affording a calculated $d_{O-H}$ of 1.15 Å (Fig. 5b). This value is approximately 0.10 Å larger than the result derived from X-ray diffraction[43].

**Dynamics of the Im guest**. The combined dielectric and diffraction measurements suggested a dynamic order–disorder transition of the Im guest in the crystal. To gain a deeper understanding of the dynamics, we applied wide-line [2]H NMR spectroscopy to the partially deuterated sample 1-d₃, which shows the same properties as 1-d₀. This spectroscopic technique can determine the geometry and frequency of rotational motions on a time-scale ranging from $10^{-8}$ to $10^{-4}$ s [47]. The experimental [2]H NMR patterns of 1-d₃ change over a large temperature range, indicating that the Im cations exhibit significant reorientation in this temperature range (Fig. 6). Due to the geometric restriction from the crystal lattice, the energetic favorable motion model of the Im cation is the in-plane motions. We adopted an in-plane two-site jump motion model (Fig. 6a and Supplementary Table 4) to explain the change in the shapes of the patterns obtained experimentally. In searching for the motion model of the Im cations, we realized that the three D atoms of 1-d₃ only give rise to one type of line shape in the patterns in Fig. 6a. This indicates that the three D atoms of 1-d₃ must undergo an identical motional modulation. Based on this requirement, we found that only the in-plane jump motion model in the intermediate regime can be a suitable candidate. Note that the motion model in this manner is somewhat similar to a restricted in-plane rotation along the axial perpendicular to the plane of Im ring. The jump angle (i.e., the reorientation angle), the rate of the jump motion, and the populations in the jump sites have strong influences on the line shape of the spectra (Supplementary Figs. 11–13). It is evident that the simulated patterns match the experimental ones quite well. The simulation reveals that the Im cations likely undergo an in-plane two-site jump motion with a jump angle of 95° (Supplementary Fig. 11). The reorientation rates at different temperatures are fit by the Arrhenius equation, $k = k_0 \exp(-E_a/RT)$, yielding an activation energy of $E_a$ = 52.1 kJ mol$^{-1}$ and a pre-exponential factor of $k_0 = 7.44 \times 10^{12}$ s$^{-1}$ (Fig. 6b). The restricted jump of the Im cation is also supported by density functional theory (DFT) calculations of the rotational potential energy profiles of 1-d₀ at 293 K (phase II) and 413 K (phase I) (Supplementary Fig. 14).

The two jump sites of the motion model may have different energy states in the crystal lattice. This may cause different populations in the sites and thus additional modulation of the line

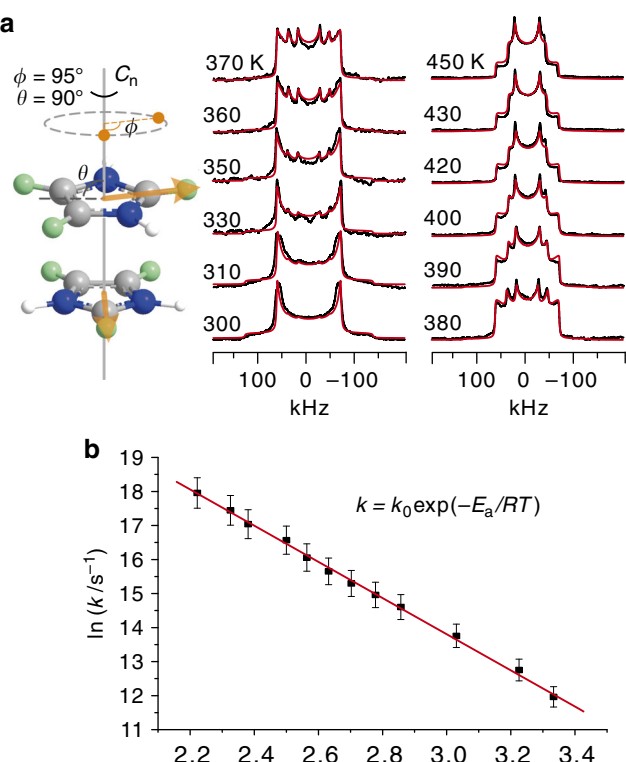

**Fig. 6** Solid-state [2]H NMR spectra and analysis. **a** Simulated (red) and experimental (black) [2]H NMR patterns of **1-d₃**. The experimental temperature varies from 300 to 450 K. The cartoon illustrates the in-plane reorientation motion of the Im cation. A two-site jump motion model is used for the [2]H pattern simulation (see the cartoon in **a**), in which $\phi$ is the jump angle, that is, the reorientation angle, and $\theta$ is the angle between the "rotation" axis $C_n$ and the visual plane containing the Im ring. The two orange points are the two jump sites of a specific deuterium atom in the visual plane. In the simulation, a rigid signal component has been added to the simulated patterns. **b** Arrhenius plot of the reorientation rates, $k$, were obtained from the simulated [2]H NMR spectra via Weblab (http://weblab.mpip-mainz.mpg.de/weblab). The error bars represent an uncertainty of 5% of the jump/reorientation rates from the pattern simulation

shape of the pattern. Supplementary Fig. 13 demonstrates the influence of the different populations in the two sites on the line shape of the pattern. Clearly, only the pattern simulated based on equal population in the two sites matches well with the features of the experimental patterns. Thus, in the simulation, we assumed the same energy state of the two sites and in turn equal population of the two sites. In most of the [2]H patterns in Fig. 6a, the outer "Pake wings", although weak, are observed in both sides of the main signals. This feature indicates the presence of a rigid signal component. To obtain a best fit between the simulated spectra and the experimental ones, we added a rigid signal component in the simulation. One possible origin of this signal component can be attributed to the temperature gradient typically present in NMR probes. In the literature, the [2]H patterns of guest molecules trapped in the meso-pores of silica materials[48] and polymer matrices[49] show complex lineshapes comparable to those in this work. The origin of the complex lineshapes is attributed to a superposition of different [2]H signal components from the domains with fast and slow molecular motion. We tried a similar approach but found that the simulated patterns did not fit the experimental ones well (Supplementary Fig. 15). Considering the characteristics of the experimental

patterns and diffraction results, we prefer to choose the motion model in the intermediate motional regime for the pattern simulation. However, note that the interpretation of the NMR results is based on the assumption that only a single crystalline phase exists in the system. In principle, the NMR results can also be interpreted by a superposition of the signals from different crystalline phases that have the same X-ray diffraction patterns but different $^2H$ NMR pattern features[50]. Based on the above X-ray and NMR observations, we cannot completely exclude this possibility.

## Discussion

The H/D isotope effect in **1** is manifested in geometric changes of the H bonds. There are two sets of H bonds in **1**: the strong O–H···O ($d_{O···O} = 2.47–2.59$ Å) bonds between the host TPA anions and the weak N–H···O ($d_{N···O} = 2.67–3.00$ Å) interactions between the guest Im cation and host TPA anions. We found that the strong H bond of the carboxyl–carboxylate pair is key to the striking GIE. Furthermore, the rigidity of the TPA unit in the host structure is essential to accumulate and transmit the secondary GIE in the H-bonded network. There are two significant consequences of the GIE: isotopic polymorphism and a downward shift of $T_{tr1}$, both of which are related with intermolecular interactions.

In comparison with the H isomorphs, the downward shift of 33 K of $T_{tr1}$ found in the D isomorphs can be accounted for by the internal pressure imposed by the secondary GIE. The small increase of approximately 0.03 Å in $d_{O···O}$ in the host causes a slight structural expansion and weakens the H-bonding interactions between the Im guest and the host (Fig. 5c). The reduced internal pressure on Im exerts a large influence on the dynamics of the confined Im cation, causing the 33 K downward shift of the $T_{tr1}$. This phenomenon has been widely found in strain-triggered phase transition materials[33], contrary to those H-bonded ferroelectrics[14, 18, 19] with upward shift of $T_{tr}$.

The isotopic polymorphism can be directly understood in terms of the internal pressure. With the elongation of $d_{O···O}$, the crystal lattice suffers from extra strain energy, which can be counterbalanced by thermal vibrations in phase II but is quickly released in phase III (phase II is only stabilized within a narrow temperature window of 14 K) via the II–III phase transition, resulting in the more stable phase III. This structural transition is witnessed by reorganizations of the supramolecular H-bonding host networks and the guest in the host channel.

In addition, the mechanism of the I–II phase transition in **1**, which is coupled with the dielectric transition, needs further clarification. In principle, this mechanism can be attributed to structural processes: (i) proton transfer in the O–H···O H bond and/or (ii) reorientation of the Im cation. However, the former can be excluded by the dielectric anisotropy. Specifically, if a proton undergoes an order–disorder transition, an induced dipole moment would appear in the direction of the H-bond chain, which should provide large $\varepsilon'$ components along the $a$ and $c$ directions; however, this was not experimentally observed (Supplementary Fig. 3). Meanwhile, the variable-temperature $^1H$ chemical shift of the O–H···O H bond in **1-d$_0$** shows only a slight upfield shift with an increase in temperature, and no sharp changes were observed during the I–II phase transition (Supplementary Fig. 10). This indicates that there is no sharp change in the O–H···O H-bond geometry within the temperature range, which is in good accordance with the temperature-dependent X-ray data. Furthermore, the downward shift of $T_{tr1}$ also excludes mechanism (i). In contrast, process (ii) can explain the large dielectric response along the $b$ direction, that is, the reorientation of the Im cation results in a net dipolar contribution to the

dielectric response (Supplementary Fig. 16). Therefore, the order–disorder transition between the frozen and motional states of the Im cations, not the proton in the H bond, is responsible for the I–II phase transition and the dielectric switching.

Finally, our study has important implications for biomolecular systems and crystal engineering. Strong H bonds are commonly found in proteins[51], such as the O–H···O bonds among the electrically charged carboxyl/carboxylate (e.g., aspartic acid and glutamic acid) residues and/or water in proteins[52, 53]. Upon deuteration, the GIE can change these H bonds and at least alter the local configurational/conformational structures that are directly/indirectly associated with physical/chemical/biological properties and functions, such as substrate binding, enzyme catalysis, protein folding and stabilization, and phase transitions and related properties. These potential effects of deuteration set a strict limit on the use of deuterated samples as proxies in structural studies of biological systems. For example, due to the low resolution of H bonds in macromolecular structures and the rarity of isotopic polymorphism, the potential significance of the GIE has been largely underestimated in neutron protein crystallography[4, 54, 55]. As a powerful tool for directly locating H/D atom positions in protein structures, this method is based on the widely accepted assumption that macromolecular crystal structures are in general not affected by deuteration. However, that is not always the case, as shown in this study. More importantly, our findings support the notion that the GIE can be used as a powerful tool for applications, such as probing and tuning H-bonded structures and related properties, understanding the nature of H bonds, improving calculations for crystal structure predictions, and designing and synthesizing new materials.

In conclusion, our study reveals a striking GIE in a hydrogen-bonded host–guest crystal. The increase in the donor–acceptor distance in the strong O–H···O hydrogen bonds in the host structure upon deuteration is responsible for the changes in the global hydrogen-bonded supramolecular structure and guest dynamics, resulting in isotopic polymorphism and a modified dielectric transition property. These findings deepen our understanding of the GIE in hydrogen bonds and suggest an approach to tune the phase-transition-related physical properties of hydrogen-bonded systems. Importantly, this work shows that a cautious consideration of the GIE must be given in related studies, such as structural identifications in biological systems.

## Methods

**Sample preparation**. All reagents in analytical grade were used without further purification. Imidazole-d$_4$ (98 atom % D) and D$_2$O (99.8 atom % D) were purchased from Sigma–Aldrich. Terephthalic acid (1.66 g, 10 mmol) and imidazole (1.36 g, 20 mmol) were dissolved in a mixed solution of DMSO (5 mL) and distilled water (10 mL). Evaporation of the solution at room temperature in air afforded block-like colorless crystals of **1-d$_0$**. Yield 65% (based on terephthalic acid). Anal. Calcd (%) for C$_{11}$H$_{10}$N$_2$O$_4$ ($M_r = 234.21$): C, 56.41; H, 4.30; N, 11.96. Found: C, 55.80; H, 4.32; N, 12.10. Selected FT-IR peaks (KBr, cm$^{-1}$): 3131 s, 1716 s. For the deuterated samples, the H/D exchanges were repeated five times via a dissolution-drying process. Because there is a phase separation in the H and D isomorphs at room temperature, the deuteration percentages of the deuterated samples are theoretically independent of the purities of the deuterated reagents and the H/D exchange times. For all measurements, large single crystals (around 500 mg) were used that were well characterized by X-ray diffractions.

**X-ray diffraction experiments**. The single-crystal X-ray diffraction data for **1-d$_0$**, **1-d$_2$**, **1-d$_3$**, and **1-d$_5$** were collected at varied temperature on a Rigaku Saturn 724$^+$ CCD diffractometer using Mo K$\alpha$ radiation ($\lambda = 0.71075$ Å) from a graphite monochromator. The structures were solved by direct methods and refined by the full-matrix method based on $F^2$ using the SHELX-97 program package[56]. The H atom in the O–H···O bond was located from the difference Fourier synthesis and isotropically refined without any restraints. It was visualized by using Platon[57]. Other H atoms were generated geometrically and refined using a riding model with $U_{iso} = 1.2U_{eq}$ (C and N). The D atoms were refined as H atoms. The

crystallographic data and selected hydrogen bonds are listed in Supplementary Tables 1 and 2.

**Thermal spectra**. TGA measurements were carried out on a TA Instruments Q500 thermogravimetric analyzer. DSC measurements were performed on a NETZSCH DSC 200F3 instrument under nitrogen at atmospheric pressure in an aluminum crucible with a heating/cooling rate of 10 K min$^{-1}$.

**Elemental analysis**. Elemental analyses of C, H, and N content were performed on a Vario MICRO analyzer.

**Dielectric spectra**. Measurements of the temperature-dependent dielectric constant were carried out on single-crystal or crystalline-powdered samples in a Ag| sample|Ag sandwich arrangement by using a Tonghui TH2828A impedance analyzer at frequencies of 1–1000 kHz with an applied electric field of 1 V.

**Solid-state NMR spectra**. The wide-line $^2$H NMR measurements were performed on a Bruker Avance III 300 spectrometer operating at 46.07 MHz for $^2$H. A Bruker two-channel static PE probe with a homemade 2.5 mm coil was used to record the $^2$H spectra. The spectra were acquired using the solid echo sequence ($90°–\tau–90°–\tau$ –acquisition). The $^2$H pulse width was 2 µs at an RF field strength of $\gamma B_1/2\pi = 125$ kHz. The refocusing delay $\tau$ was approximately 28 µs.

**Theoretical calculations**. DFT calculations were performed using the Vienna ab initio simulation package[58]. The projector augmented wave method was used to describe the interactions between ions and electrons[59, 60]. The exchange–correlation interaction functional was the generalized gradient approximation in the Perdew–Burke–Ernzerhof functional[61]. The van der Waals interactions were included by using Grimme's dispersion-corrected semi-empirical DFT-D2 method[62]. The cutoff energy for the plane wave basis set was set to 400 eV. The integration over the irreducible Brillouin zone was carried out over a 7 × 3 × 5 Monkhorst–Pack grid[63]. A single $\Gamma$-point calculation was sufficient for sampling the Brillouin zone. The potential-energy curve for the rotation of the Im cation in **1-d$_0$** was plotted by calculating single-point energies every 10° through a 360° rotation. The rotational axis was defined to be perpendicular to the Im ion plane and across the center of the five-membered ring.

**Data availability**. The X-ray crystallographic data reported in this article were deposited at the Cambridge Crystallographic Data Centre (CCDC), under deposition numbers CCDC 1493602–1493622. These data can be obtained free of charge from The Cambridge Crystallographic Data Centre via www.ccdc.cam.ac. uk/data_request/cif. All other data that support the findings of this study are available from the corresponding author on request.

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

## Acknowledgements

This work was financially supported by National Natural Science Foundation of China (21225102 and 21574043), National Key Basic Research Program of China (973 Program) (2013CB921801), and National High-Tech R&D Program of China (863 Program) (2014AA123400 and 2014AA123401). We thank the staffs from BL17B beamline of National Facility for Protein Science in Shanghai (NFPS) at Shanghai Synchrotron Radiation Facility, for assistance during data collection.

## Author contributions

W.Z. conceived the project and designed the experiments. C.S. and C.-H.Y. performed synthesis and characterization experiments. X.Z. and Y.-F.Y. measured the solid-state NMR spectra. W.Z. and Y.-F.Y. analyzed the data and wrote the manuscript. All authors discussed the results and commented on the manuscript.

## Additional information

**Competing interests:** The authors declare no competing financial interests.

