## [Peer Review File · Nature Communications]

Reviewers' comments:

Reviewer #1 (Remarks to the Author):

This is interesting and well-conducted research, although the hype could be reduced. This statement:

"However, the GIE on global structures and bulk properties of solid-state materials has been scarcely explored."

...immediately put me in a bad mood; there are literally dozens of such cases. The authors should check out the effect of deuterium labelling on H₂SO₄, and pyridine, which are of comparable magnitude. More are referenced in our 2005 paper.

However, the rest of the paper is excellent, and its significance is still substantial despite the unnecessary hype. It is comprehensive, including crystal structures of all phases and isotope substitution patterns, full DSC data, and some NMR (which corroborates the crystallography nicely)

I was a little worried by the size of the preexponential factor for the reorientation, because it's huge, and the data seem to have been collected in a single set ignoring the phase transition. There seems to be a substantial discontinuity in the ²H NMR spectra between 370 K and 390 K. Moreover, the Arrhenius plot in 6(b) omits the data collected between 290K and 250K and simulated in 6(a); why? Are they discrepant?

It's an interesting and likely significant system, and aside from the above, relatively minor reservations, which should be attended to, I think it could be accepted following minor revision.

Reviewer #2 (Remarks to the Author):

Report on manuscript

Geometric isotope effect of deuteration in a hydrogen-bonded host-guest crystal

by

Chao Shi, Xi Zhang, Chun-Hua Yu, Ye-Feng Yao,, Wen Zhang,

Submitted to Nature Communications

This article discusses a very important subject of hydrogen-bonded materials and their exceptional properties. In this case these are phase transitions involving the hydrogen-bonds transformations and the isotope effect of H/D substitution. Authors chose the imidazole:terephthalate [Im]+[TPA]- co-crystal, where the following H/D substitutions have been studied: (1) not at all; (2) of two acidic protons; (3) of 3 C-H atoms in Im; and (4) of both these (2+3) substitutions.

Within the studied range these compounds undergo two or three phase transitions that Authors describe as the transitions between LDS and HDS (low- and high-dielectric states).

In my opinion there are no LDS-HDS termed phase transitions (this term appeared recently in the chemical literature, but not in the specialist literature on phase transitions) – in fact all phase transitions change the electric permittivity to some extent, so all phase transitions are between LDS and HDS. Authors say that in their case the permittivity increases 6 times, which is nothing exceptional at all (not to mention the transitions increasing permittivity 1000 times). For these reasons phase transitions are not classified according to the permittivity, but there are well established

methods. In this area Authors fail totally – they do not say what type of phase transitions they detected in their materials – 1st or 2nd order – which is essential for some of their analysis included in the text. They also use term 'paraelectric' for one of the phases, while it is reserved for ferroelectric materials above T_c . Of course the problem is that the [Im]+[TPA]- co-crystal is not ferroelectric (according the text all phases are centrosymmetric – no studies for the possible anti-ferroelectricity were made, either). Then Authors apply the Curie-Weiss formalism, which is applicable to the paraelectric phase of ferroelectric crystals. To me the phase transitions to phases I of compounds 1-do and 1-d5 appear as the 2nd order ones (as I judge from the DSC in Figure 2), but then parameter T_0 of the Curie Weiss law should be equal to T_{tr} , which apparently is nearly 100K different. I could go on pointing out similar important errors in the thermodynamics of the phase transitions, but it is most important that this study does not reveal new information on the H/D isotope effects: it is well known that the phase transitions can be shifted, up or down depending on the deuteration site – exactly as it is in the [Im]+[TPA]- co-crystal; it also happens that some of the phases disappear after the deuteration. Unfortunately Authors fail to refer to this rich literature.

For these reasons I cannot recommend this paper for publication.

Reviewer #3 (Remarks to the Author):

This is a very interesting manuscript reporting unusual isotope effects on the hydrogen bond geometries and dielectric properties of a crystalline hydrogen-bonded host-guest crystal, imidazolium hydrogen terephthalate (1). The system reveals three different phases which interconvert upon heating. X-ray crystallographic studies show that OHN hydrogen bonds of moderate strength link the guest to the host, and that the host is held together by OHO hydrogen bonds between the homoconjugate terephthalate carboxylic acid anions. 1 exhibits an order-disorder phase transition around 400 K associated with proton transfer in the OHO hydrogen bonds and with a strong increase of the dielectric constant. Whereas the O...O distances before H-bond deuteration exhibit only a small temperature dependence, after H-bond deuteration the O...O distances have strongly increased in phase 1 at low temperatures. They decrease again in phases 2 and 3. The phase transition temperature between the latter is smaller for the deuterated system. ^2H NMR studies of imidazolium deuterated in the carbon positions indicate a molecular mobility leading to 95° two-site rotations within the molecular plane. ^1H NMR experiments reveal also the presence of strong OHO hydrogen bonds.

I recommend publication of the results in Nature Communications, but I think the authors should perform some minor changes in order to attract readers which are less familiar with this field.

(1) In the beginning, the authors should describe the term "isotopic polymorphism". To my knowledge, this term refers to a different crystal structure of a given system after partial or full deuteration, here associated to different hydrogen bonded pathways. That is the case here in a small temperature interval where the protonated system forms phase 2 and the deuterated phase 1.

(2) In the legend of Figure 1 the difference between "isotopolog" and "isomorph" is not clear to me. Why does one need the latter term? Please define the different terms.

(3) I do not understand the spheres in the center of the aromatic rings. Please comment.

(4) I have difficulties to see colors, therefore I have problems to understand Figures 4 and 5. For me, the H-bond pathways are better understood by the graphs on the right side of Figure 5.

On the other hand, in Figure 5 I have difficulties with the color encoding of the data points in the left side. Why 1-d5 gives rise to two curves? Also, the color of the texts with the distances in the right side of Figure 5 refer to different isotopologs, but please add to which one.

(5) The authors write " Upon deuteration, significant geometric changes of the H-bonds and the host structure modulate the dynamics and arrangement of the Im guest in the host channel, leading to the occurrence of unconventional downward shift of the Ttr and a new phase (known as isotopic polymorphism)." Please state the numbering of the phases before and after deuteration.

(6) "We next introduced an in-plane two-site jump motion model". Please add a scheme characterizing the initial and final state of the jump. The inserted graph in Figure 6a is not clear enough. Discuss whether both sites exhibit the same energy with respect to the crystal frame (equilibrium constant equal 1), or whether one has to expect an equilibrium constant unequal 1. State whether a superposition of domains with fast and with slow exchange can be excluded (see for comparison Fig. 5 in [dx.doi.org/10.1021/jp012391p](https://doi.org/10.1021/jp012391p))

Reviewers' comments:

Reviewer #1 (Remarks to the Author):

This is interesting and well-conducted research, although the hype could be reduced.

This statement:

"However, the GIE on global structures and bulk properties of solid-state materials has been scarcely explored."

...immediately put me in a bad mood; there are literally dozens of such cases. The authors should check out the effect of deuterium labelling on H₂SO₄, and pyridine, which are of comparable magnitude. More are referenced in our 2005 paper.

Reply:

Thanks for this comment!

We revised this sentence in the abstract section as

"To understand the GIE on global structures and bulk properties is of great importance for the study of structure-property relationship of hydrogen-bonded systems."

Moreover, we revised the introduction section of the manuscript to highlight the GIE in some reported hydrogen-bonded structures, such as H₂SO₄, and pyridine. Additional references (ref 8 and 9) were added.

However, the rest of the paper is excellent, and its significance is still substantial despite the unnecessary hype. It is comprehensive, including crystal structures of all phases and isotope substitution patterns, full DSC data, and some NMR (which corroborates the crystallography nicely).

I was a little worried by the size of the preexponential factor for the reorientation, because it's huge, and the data seem to have been collected in a single set ignoring the

phase transition. There seems to be a substantial discontinuity in the ^2H NMR spectra between 370 K and 390 K. Moreover, the Arrhenius plot in 6(b) omits the data collected between 290 K and 250 K and simulated in 6(a); why? Are they discrepant?

Reply:

We re-simulated all of the patterns again using the same motion model but adding an additional rigid signal component. The origin of this rigid signal component can be attributed to the temperature gradient typically present in NMR probe, or the presence of the second phase/domain where the motions of the Im cations are highly restricted even at the high temperatures. The new simulation fits the patterns better than the old one and yields an activation energy of 52.1 kJ/mol and the pre-exponential factor of $7.44 * 10^{12} \text{ s}^{-1}$.

The line shape between 370 K and 390 K shows significant change. But based on our model, this is not a discontinuity, but rather a continuous change upon the jump rate. For a better explanation of this point, we updated Figure S12 by extending the range of the jump rate.

In the old Figure 6b, we omitted the data between 300 K and 350 K by occasional inattention. In the new version, we made the simulation for all of the data acquired between 300 K and 450 K. No discrepancy is observed in the curve.

It's an interesting and likely significant system, and aside from the above, relatively minor reservations, which should be attended to, I think it could be accepted following minor revision.

Reviewer #2 (Remarks to the Author):

Report on manuscript

Geometric isotope effect of deuteration in a hydrogen-bonded host–guest crystal

by

Chao Shi, Xi Zhang, Chun-Hua Yu, Ye-Feng Yao., Wen Zhang,

Submitted to Nature Communications

This article discusses a very important subject of hydrogen-bonded materials and their exceptional properties. In this case these are phase transitions involving the hydrogen-bonds transformations and the isotope effect of H/D substitution. Authors chose the imidazole:terephthalate $[\text{Im}]^+[\text{TPA}]^-$ co-crystal, where the following H/D substitutions have been studied: (1) not at all; (2) of two acidic protons; (3) of 3 C–H atoms in Im; and (4) of both these (2+3) substitutions.

Within the studied range these compounds undergo two or three phase transitions that Authors describe as the transitions between LDS and HDS (low- and high-dielectric states). In my opinion there are no LDS-HDS termed phase transitions (this term appeared recently in the chemical literature, but not in the specialist literature on phase transitions) – in fact all phase transitions change the electric permittivity to some extent, so all phase transitions are between LDS and HDS. Authors say that in their case the permittivity increases 6 times, which is nothing exceptional at all (not to mention the transitions increasing permittivity 1000 times).

Reply:

We would like to clarify that we never term the phase transitions by LDS-HDS. In the introduction section, we pointed out that the switchable dielectric constant or dielectric transition is a phase transition-associated bulk property. It indeed does not refer to phase transition but a bulk switching property of dielectric constant, corresponding to a type of responsive dielectrics. We are sorry for this misleading

probably due to the very brief introduction of the newly reported term in the manuscript. In the revised manuscript, we revised the related description as

"Our work focuses on molecule-based dielectrics that undergo dielectric transitions between relatively low- and high-dielectric states³⁰⁻³², a type of recently identified switchable dielectrics as the electrical counterpart to magnetic spin crossover materials^{33,34}. The T_{tr} , as a key parameter to describe the dielectric switching temperature, is determined by motional changes of polar molecules between orientationally ordered (frozen) and dynamically disordered (melt-like) states in the crystal lattice³¹."

In order to compare the difference of the two dielectric states, we would like to keep the terms "LDS and HDS (low- and high-dielectric states)".

It is true that this term (switchable dielectric constant or dielectric transition) has been only recently raised in chemical literatures, but not in the specialist literature on phase transitions. In fact, it is firstly used by us since 2010 in series of papers, including: (1) Zhang, W., Cai, Y., Xiong, R.-G., Yoshikawa, H., Awaga, K. Exceptional dielectric phase transitions in a perovskite-type cage compound. *Angew. Chem. Int. Ed.* **49**, 6608–6610 (2010); (2) Zhang, W., Ye, H.-Y., Graf, R., Spiess, H. W., Yao, Y.-F., Zhu, R.-Q., Xiong, R.-G. Tunable and switchable dielectric constant in an amphidynamic crystal. *J. Am. Chem. Soc.* **135**, 5230–5233 (2013); (3) Shi, C., Yu, C.-H., Zhang, W. Predicting and screening dielectric transitions in a series of hybrid organic–inorganic double perovskites via an extended tolerance factor approach. *Angew. Chem. Int. Ed.* **55**, 5798–5802 (2016).

In the Reviewer's opinion, "in fact all phase transitions change the electric permittivity to some extent, so all phase transitions are between LDS and HDS." However, we herein would like to point out some distinct characteristics of dielectric transition (we uses this term hereafter) and its implications for responsive materials.

(1) Firstly, we use dielectric transition to describe a type of molecular dielectrics with distinct changes of dielectric constant due to motional changes of molecular dipoles.

The significance of this property is that it is an electrical counterpart to the magnetic spin crossover or spin transition (see Figure R1). Both of spin transition and dielectric transition are the properties mostly relying on design and synthesis of molecules that chemists are familiar with.

(a)

(b)

Figure R1. (a) One-to-one correspondence between magnetic and dielectric properties; (b) Analogies and differences between dielectric transition and spin transition (taking Fe(III) for example).

(2) Secondly, it is not the case that "so all phase transitions are between LDS and HDS". According to Figure R1, the dielectric transition described in our manuscript

specifically refers to molecular systems that can show a transition between two dielectric states, i.e., static/ordered and dynamic/disordered states, corresponding to low- and high-dielectric states. Its microscopic structural origin comes from motional changes of the non-interacting polar components which lead to macroscopic polarization changes. The property could lead to a type of responsive dielectric materials that have potential applications in sensing, switching and energy transformation. Note: Ferroelectricity would emerge if the dipoles become coupled.

For dielectrics that have no dipolar orientations, the dielectric changes upon phase transitions would be complicated. Some undergoing ferroelectric transition or ferroelectric relaxation show very high dielectric changes as the Reviewer pointed out that "the transitions increasing permittivity 1000 time". Some others undergoing slight structural reorganizations only show very small dielectric changes (e.g., less than 1 time) due to highly restricted internal dipolar motions or structural distortion-induced polarization changes.

(3) Thirdly, the Reviewer pointed out that "Authors say that in their case the permittivity increases 6 times, which is nothing exceptional at all". This value is moderate in molecule-based dielectric transitions. Whether it is exceptional or not is beyond the central topic of this manuscript.

For these reasons phase transitions are not classified according the permittivity, but there are well established methods. In this area Authors fail totally – they do not say what type of phase transitions they detected in their materials – 1st or 2nd order – which is essential for some of their analysis included in the text.

Reply:

We would like to clarify that we did not intend to classify the phase transitions according to the permittivity.

In order to clarify what type of the phase transitions from a thermal dynamic

viewpoint, we performed scanning rate dependence of the DSC measurements on samples 1-d₁ and 1-d₅ and drew normalized unit cell volume of **1-d₅** versus temperature plot. Two new figures were added in SI as Figure S2b and S2c. A sentence was added in the "Synthesis and phase transitions" subsection as

"The I-II and II-III phase transitions in **1-d₅** show characteristics of second and first order, respectively (Fig. S2b and S2c). "

They also use term 'paraelectric' for one of the phases, while it is reserved for ferroelectric materials above T_c. Of course the problem is that the [Im]⁺[TPA]⁻ co-crystal is not ferroelectric (according the text all phases are centrosymmetric – no studies for the possible anti-ferroelectricity were made, either). Then Authors apply the Curie-Weiss formalism, which is applicable to the paraelectric phase of ferroelectric crystals. To me the phase transitions to phases I of compounds 1-d₀ and 1-d₅ appear as the 2nd order ones (as I judge from the DSC in Figure 2), but then parameter T₀ of the Curie Weiss law should be equal to T_{tr}, which apparently is nearly 100K different.

Reply:

Thanks for this comment.

The reviewer thinks the term paraelectric is "reserved for ferroelectric materials above T_c". In our opinion, the term would be more widely used if only there are free or non-interacting dipolar orientations in the crystals. This is the case for spin transition, that is, in the high-spin state, the system can be seen as a paramagnetic phase. For example, the spin transition can be thought as a paramagnetic-diamagnetic transition in Fe(II) spin crossover compounds. However, in order to avoid any potential misuse and misleading, we deleted this description in the revised manuscript and strictly followed the Reviewer's definition.

As to the use of Curie-Weiss law, we admit its use would be inappropriate, "which is

applicable to the paraelectric phase of ferroelectric crystals" as the Reviewer pointed out. Therefore, we deleted the corresponding descriptions of the law in the revised manuscript. The discrepancy between the Curie-Weiss constant T_0 and DSC result was thus resolved.

The possibility of the ferroelectricity or antiferroelectricity of **1** was excluded base on present studies. Experimental proofs include:

(1) No change of response of second-harmonic generation (SHG) of **1** at T_{tr} indicates the crystal always adopts centrosymmetric space group (Figure R2). This measurement excludes the possibility of ferroelectricity.

Figure R2. Temperature dependence of SHG signal of **1-d₀**. For a ferroelectric-paraelectric transition, there is a SHG change at the T_{tr} .

(2) Preliminary polarization-electric field measurement does not show characteristic double loops of antiferroelectricity (Figure R3).

Figure R3. Polarization versus voltage plot of $1-d_0$. The applied electric field is about 20 kV/cm.

I could go on pointing out similar important errors in the thermodynamics of the phase transitions, but it is most important that this study does not reveal new information on the H/D isotope effects: it is well known that the phase transitions can be shifted, up or down depending on the deuteration site – exactly as it is in the $[Im]^+[TPA]^-$ co-crystal; it also happens that some of the phases disappear after the deuteration. Unfortunately Authors fail to refer to this rich literature.

Reply:

Our focus of the system is mainly on the GIE effect on the hydrogen bonds, phases and isotopic polymorphism, and phase transition-driven dielectric transition. It can be seen that the main content of the manuscript is the structural investigation of the GIE, including first and second GIE and the structural origin of the isotopic polymorphism. Our study, in particular, reveals the relationship and transformation among the polymorphs under external stimuli (herein, temperature), which has been rarely investigated. In order to make this background clearer, we revised the introduction section of the manuscript. References on the H/D isotope effect on the phase transition behaviors were properly cited as refs 12-28.

For these reasons I cannot recommend this paper for publication.

Reply:

We carefully fixed the errors and revised the manuscript with the emphasis on geometric isotope effect on structural phases. We hope these revisions would improve the quality of the manuscript and make it acceptable.

Reviewer #3 (Remarks to the Author):

This is a very interesting manuscript reporting unusual isotope effects on the hydrogen bond geometries and dielectric properties of a crystalline hydrogen-bonded host-guest crystal, imidazolium hydrogen terephthalate (**1**). The system reveals three different phases which interconvert upon heating. X-ray crystallographic studies show that OHN hydrogen bonds of moderate strength link the guest to the host, and that the host is held together by OHO hydrogen bonds between the homoconjugate terephthalate carboxylic acid anions. **1** exhibits an order-disorder phase transition around 400 K associated with proton transfer in the OHO hydrogen bonds with a strong increase of the dielectric constant. Whereas the O...O distances before H-bond deuteration exhibit only a small temperature dependence, after H-bond deuteration the O...O distances have strongly increased in phase 1 at low temperatures. They decrease again in phases 2 and 3. The phase transition temperature between the latter is smaller for the deuterated system. ^2H NMR studies of imidazolium deuterated in the carbon positions indicate a molecular mobility leading to 95° two-site rotations within the molecular plane. ^1H NMR experiments reveal also the presence of strong OHO hydrogen bonds.

I recommend publication of the results in Nature Communications, but I think the authors should perform some minor changes in order to attract readers which are less

familiar with this field.

(1) In the beginning, the authors should describe the term "isotopic polymorphism". To my knowledge, this term refers to a different crystal structure of a given system after partial or full deuteration, here associated to different hydrogen bonded pathways. That is the case here in a small temperature interval where the protonated system forms phase 2 and the deuterated phase 1.

Reply:

We added a paragraph to describe the term "isotopic polymorphism" in the introduction section, which balanced the focuses of the manuscript.

"A long-held assumption is that deuteration seems generally not to alter the crystal structures⁴. It is true in most cases; but there emerge several exceptions, which unjustifies this assumption, that show different crystal structures upon deuteration (a phenomenon known as "isotopic polymorphism")⁵, including oxalic acid dihydrate, complex of pentachlorophenol and 4-methylpyridine, pyridine, and so on⁵⁻⁷. Occurrence of the isotopic polymorphism in the crystals seems serendipitous and unpredictable⁸, which, however, keeps the clues to understand the interplays among intermolecular interactions in crystal packing (a central issue in crystal engineering). Hydrogen (H) bonds, by treated as supramolecular synthons⁹, play a vital role in crystal engineering and become a more predictable approach to exhibit the isotopic polymorphism. This is because deuteration of the H bond can cause geometric changes in the bond, known as geometric H/D isotope effect (GIE)^{10,11}, and this effect would accumulate and transmit into the whole crystal phase."

(2) In the legend of Figure 1 the difference between "isotopolog" and "isomorph" is not clear to me. Why does one need the latter term? Please define the different terms.

Reply:

In chemistry, isotopologs refer to any of a group of compounds differing in the number, and/or position of substitution of atoms by isotopes while isomorphs refer to

crystals belonging to the same space group.

We defined the two terms in the legend of Figure 1.

(3) I do not understand the spheres in the center of the aromatic rings. Please comment.

Reply:

The spheres are dummy atoms that are used to depict $\pi \cdots \pi$ interaction between two phenyl rings.

In order to avoid this confusion, we deleted the spheres and redrew Figure 3b. A new Table S3 was added in SI in which details of the $\pi \cdots \pi$ interactions in **1-d₀** and **1-d₅** were summarized.

(4) I have difficulties to see colors, therefore I have problems to understand Figures 4 and 5. For me, the H-bond pathways are better understood by the graphs on the right side of Figure 5.

On the other hand, in Figure 5, I have difficulties with the color encoding of the data points in the left side. Why **1-d₅** gives rise to two curves? Also, the color of the texts with the distances in the right side of Figure 5 refer to different isotopologs, but please add to which one.

Reply:

In order to make the figures more readable, we added a note to define the phase colors: (phase I, light pink; phase II, yellow; phase III, light cyan).

The reason that **1-d₅** gives rise to two curves in phase III is that there emerge two types of OHO hydrogen bonds between the TPA cations because of the phase transition from II to III. This point was explained in the sentence in the subsection of

GIE on H bonds and supramolecular structures as

"Furthermore, the II–III phase transition of **1-d₅** shows a jump in $d_{O...O}$ to approximately 2.550/2.587 Å (there emerge two types of $d_{O...O}$ in phase III of the D isomorphs), with an increase of approximately 0.100 Å compared with that of **1-d₀**, resulting in the occurrence of the isotopic polymorphism."

The colors of the texts with the distances in the right side of Figure 5 refer to different isotopologs. We added a note to define them: (black for **1-d₀** and blue for **1-d₅**)

(5) The authors write " Upon deuteration, significant geometric changes of the H-bonds and the host structure modulate the dynamics and arrangement of the Im guest in the host channel, leading to the occurrence of unconventional downward shift of the Ttr and a new phase (known as isotopic polymorphism)." Please state the numbering of the phases before and after deuteration.

Reply:

We stated the numbering of the phases before and after deuteration in the first sentence of the last paragraph in the introduction section.

"Herein, we establish an H-bonded host-guest model system, [Im][TPA] (**1**; Im = imidazolium cation, TPA = hydrogen terephthalate monoanion), to demonstrate significant GIE on the H bonds, the phases (two and three phases before and after deuteration, respectively) and the bulk dielectric transition properties under variable temperature conditions."

(6) "We next introduced an in-plane two-site jump motion model". Please add a scheme characterizing the initial and final state of the jump. The inserted graph in Figure 6a is not clear enough. Discuss whether both sites exhibit the same energy with respect to the crystal frame (equilibrium constant equal 1), or whether one has to expect an equilibrium constant unequal 1. State whether a superposition of domains

with fast and with slow exchange can be excluded (see for comparison Fig. 5 in [dx.doi.org/10.1021/jp012391p](https://doi.org/10.1021/jp012391p))

Reply:

We scrutinized the raised points. We think that the two sites should have the same energy. If not, the populations in the two sites most likely will be different. The different populations in the two sites will give significant influence on the line shape of ^2H patterns. We added the simulated patterns in Supporting Information showing this influence (see Figure S13). It is clear that only the pattern simulated based on the equal population in the two sites matches well with the features of the experimental patterns. Based on this observation, we thus assumed the same energy and population of the two sites. To clarify this point, the following texts were added in the main text:

“The two jump sites of the motion model may have different energy states in the crystal frame. This may cause the different populations in the sites and thus the additional modulation on the line shape of pattern. Fig. S13 demonstrates the influence of the different populations in the two sites on the line shape of the pattern. It is clear that only the pattern simulated based on the equal population in the two sites matches well with the features of the experimental patterns. In the simulation, we thus assumed the same energy state of the two sites and in turn the equal population in the two sites.”

We checked the experimental patterns carefully and realized the presence of a rigid signal component in most of the patterns. The weak “Pake wings” in both sides of the main signals evidence the presence of this signal component. We thus added this signal component in the simulation. The origin of this component could be due to the temperature gradient typically present in NMR probes, or the presence of the domains where the Im cations have a very restricted dynamics even at the high temperatures. To clarify this point, the following texts have been added in the main text:

“In most of the ^2H patterns in Fig. 6a, the outer “Pake wings”, although weak, are observed in both sides of the main signals. This feature indicates the presence of the rigid signal component. To have a best fit between the simulated spectra and the

experimental ones, we have add a rigid signal component in the simulation. The origin of this signal component most likely is due to the temperature gradient typically present in NMR probes. But we cannot exclude the possibility that the rigid signal component could also originate from the domains where the Im cations have a very restricted dynamics even at the high temperatures⁴⁷.”

Reviewers' comments:

Reviewer #1 (Remarks to the Author):

The authors have adequately responded to my concerns in their revised m/s. However, I'm not really qualified to evaluate the dielectric measurements.

Reviewer #3 (Remarks to the Author):

The authors have considerably improved their paper. Almost all comments I raised have been taken into account. Only with the answer to (6) I am not yet happy.

I had asked the authors to "state whether a superposition of domains with fast and with slow exchange can be excluded (see for comparison Fig. 5 in [dx.doi.org/10.1021/jp012391p](https://doi.org/10.1021/jp012391p)), but I could not find a satisfying answer to that point. For the sake of the authors, they really should check whether it is possible to simulate the 2H NMR spectra using a superposition of two spectral components arising from rigid domains and from domains with fast motions. I have seen many solids exhibiting phase transitions of that type. The authors should perform the "Roessler" analysis (see [dx.doi.org/10.1021/jp012391p](https://doi.org/10.1021/jp012391p) and papers cited therein) where the fraction of the fast and slow components are plotted as a function of temperature. From that curve the average energy of activation as well as the width of the distribution of activation energies can be derived. The average activation energy should then compared with those obtained by lineshape analysis.

The latter should be very carefully checked again. I was shocked to learn that adding simply a rigid spectral component in the simulation reduces the pre-exponential factor from $6.13 \times 10^{18} \text{ s}^{-1}$ to $7.44 \times 10^{12} \text{ s}^{-1}$. That means that values of the activation energy and of the pre-exponential factor may be strongly affected by systematic errors. It is, therefore, not appropriate to publish so exact numbers which may change strongly depending on the way of how the analysis is done.

I hope that the authors will take this comment into account, so that I will be able to recommend publication of a suitably revised manuscript.

Reviewer #4 (Remarks to the Author):

The manuscript reports the novel H/D effect for the structural transition of host-guest crystals. The author examined the deuterated effects of the dielectric phase transition for molecular motion of Im cation, where the D replacement expanded the host hydrogen-bonding lattice. The essential point of the inverse D/H effect is the unusual host lattice transformation coupled with the motional freedom of cation. Although such phenome should be one of the interesting phase transition mechanism, the motional freedom of Im cation has not been clearly discussed as pointed out by the reviewer. The detail simulation results of NMR measurements for each D isomer should be carefully mentioned, and should be reviewed by specialist of solid state NMR. In minor point, the standard deviation should be included in hydrogen-bonding distances. After the minor revision above, the paper should be progressed in the publication of Nature Comm.

Reviewers' comments:

Reviewer #1 (Remarks to the Author):

The authors have adequately responded to my concerns in their revised m/s. However, I'm not really qualified to evaluate the dielectric measurements.

Reply:

In the revised manuscript, we added a new reference:

17. Shi, C., Han, X.-B., Zhang, W. Structural phase transition-associated dielectric transition and ferroelectricity in coordination compounds. *Coord. Chem. Rev.* <https://doi.org/10.1016/j.ccr.2017.09.020> (2017).

In this review article, we summarized recent achievements on two types of structural phase transition-associated properties, i.e., dielectric transition and ferroelectricity, in coordination compounds. In particular, the basic concepts and fundamentals of the dielectric transition property are introduced. We expect this would help the readers to gain a more comprehensive understanding of the structural phase transition associated bulk dielectric transition.

Reviewer #3 (Remarks to the Author):

The authors have considerably improved their paper. Almost all comments I raised have been taken into account. Only with the answer to (6) I am not yet happy.

I had asked the authors to "state whether a superposition of domains with fast and with slow exchange can be excluded (see for comparison Fig. 5 in [dx.doi.org/10.1021/jp012391p](https://doi.org/10.1021/jp012391p)), but I could not find a satisfying answer to that point.

For the sake of the authors, they really should check whether it is possible to simulate the ^2H NMR spectra using a superposition of two spectral components arising from rigid domains and from domains with fast motions. I have seen many solids exhibiting phase transitions of that type. The authors should perform the "Roessler" analysis (see [dx.doi.org/10.1021/jp012391p](https://doi.org/10.1021/jp012391p) and papers cited therein) where the fraction of the fast and slow components are plotted as a function of temperature. From that curve the average energy of activation as well as the width of the distribution of activation energies can be derived. The average activation energy should then compared with those obtained by lineshape analysis. The latter should be very carefully checked again.

Reply:

Thanks for this comment!

We have considered carefully the motion models combining the rigid component and the component from the fast motion. However, after many attempts, we cannot find a

feasible fast motion model having clear physical meaning which can be used to well reproduce the patterns. The tested motion models and the reasoning are shown below.

First, we would like to give a short discussion about the molecular features of the partially deuterated imidazolium- \mathbf{d}_3 cation in $\mathbf{1-d}_3$. Based on such knowledge, the motion models that have been tried in the pattern simulation are discussed.

Figure 1. The molecular geometry of imidazolium- \mathbf{d}_3 in $\mathbf{1-d}_3$.

The molecular features of $\mathbf{1-d}_3$:

The cartoon picture in Fig. 1 describes the molecular features of imidazolium- \mathbf{d}_3 .

According to the molecular nature, the line a, b, c are coplanar. The angles between line a and b, a and c, are equal (144°). The angle between line b and c is 72° . θ_i ($i = 1, 2, 3$) describes the polar angle between line a/b/c and z axis. In general, $\theta_1 \neq \theta_2 \neq \theta_3$.

But there are several special situations:

Case 1. When z axis is perpendicular to the imidazolium ring, $\theta_1 = \theta_2 = \theta_3$;

Case 2. When z axis is in the angular bisector plane of the angle between the two lines of line a, b, c, $\theta_m = \theta_n \neq \theta_l$ ($m, n, l = 1, 2, 3$). For example, if z axis is in the

angular bisector plane of the angle between the line b and c, $\theta_2 = \theta_3 \neq \theta_1$.

Case 3. When z axis is in the plane that is perpendicular to the angular bisector plane of the angle between the two lines of line a, b, c, $\theta_m = \theta_n \neq \theta_1$ ($m, n, l = 1, 2, 3$). For example, if z axis is in the plane that is perpendicular to the angular bisector plane of the angle between the line b and c, $\theta_2 = \theta_3 \neq \theta_1$.

The fast motion model: non-in-plane motion & in-plane motion (rotation, jump motion and vibration)

1. Non-in-plane motion

In principle, this motion will result in three different signal components which are corresponding to the three different polar angles, i.e., $\theta_1 \neq \theta_2 \neq \theta_3$. In the experimental patterns acquired between 350 K and 400 K, the pattern lineshape indicates three signal components, that is, the rigid component (the outer horns) and the two other components (the two sets of inner horns). If the two other signal components (the two sets of inner horns having the splitting of 46.1 kHz and 81.2 kHz, respectively) are from the fast motion, this means that the motion geometry must fulfill the requirement described in Case 2 or 3 above. And more restrictedly, the polar angles, $\theta_m = \theta_n \neq \theta_1$, have be chosen to the specific values that can yield the two quadrupolar splitting of 46.1 kHz and 81.2 kHz simultaneously. We have tried the rotation, 2-site jump, and local wobbling with the z axis described in Case 2 or 3, but cannot find the proper motion model to reproduce the simulated

patterns close to the experimental ones. The non-in-plane motion is thus excluded.

2. In-plane motion

In-plane axial rotation

In this case, z axis is perpendicular to the imidazolium ring, $\theta_1 = \theta_2 = \theta_3$. This means that the three deuterium spins of one imidazolium-d₃ cation only produce one signal component. We may still focus on the experimental patterns acquired between 350 K and 400 K. If the two sets of inner horns in the patterns are from the fast motions, two fast motion models are required. To obtain the two quadrupolar splitting of 46.1 kHz and 81.2 kHz, the simple axial rotation can be excluded, because it yields a quadrupolar splitting of 67 kHz (The quadrupolar splitting of 134 kHz is assumed for N–D in a static case in this study). The simulated patterns are shown in Figure 2.

Figure 2. The in-plane axial rotation model and the simulated patterns based on the

static model and the rotation model.

In-plane 2-site jump motion

The 2-site jump motion model may reproduce the required splitting (Figure 3).

The following are the simulated patterns based on the 2-site jump motion with different jump angles. Based on this 2-site jump motion model, a jump motion with the jump angles of 55.5° , 84.3° , 95.7° and 124.5° can reproduce the patterns with the distance of 46.1 kHz between two horns, while a jump motion with the jump angles of 42.1° and 137.9° can reproduce the patterns with the distance of 81.2 kHz between two horns.

Figure 3. The 2-site jump motion model and the simulated patterns based on this motion model with the different jump angles.

In order to have the two sets of horns in the pattern, the patterns from the jump motions with two jump angles (e.g., 55.5° and 42.1°) need to be summed up. The summed patterns and the component ratios are shown in Figure 4. Comparison shows that the pattern having the component ratio (A:B) of 45:55 has the best fit to the experimental ones.

Figure 4. (Left) The simulated patterns based on the jump motion model with the jump angle of 55.5° and 42.1° . (Right) The patterns obtained by summing the two patterns simulated based on the jump motion model with the jump angles of 55.5° and 42.1° . The component ratios, A:B, are shown in the figure.

Figure 5. (a) The procedure to obtain the patterns in (b). In the first summation, the two patterns are simulated based on the jump motion model in the fast limit with the jump angles of 55.5° and 42.1° . The component ratio A:B is 45:55. (b) The summed patterns. (c) The comparison between the experimental patterns acquired at 350 K and 370 K (the black lines) and the simulated patterns.

In order to add the rigid component, the patterns simulated based on the jump motion model need to sum with the pattern simulated from the static model. Figure 5a shows the procedure from which the patterns in Figure 5b can be obtained. The patterns in Figure 5b demonstrate the lineshape change upon the change in the content of the rigid component. In Figure 5c, the experimental patterns acquired at 350 K and 370 K are compared with the simulated patterns. It is clear that the simulated patterns do not fit the experimental ones (The inner horns in the simulated patterns are significantly wider than those in the experimental patterns).

We also simulated the patterns based on the similar in-plane jump motion in the fast limit but with the different jump angles, i.e., (55.5° , 84.3° , 95.7° and 124.5°) and (42.1° and 137.9°). But the results show that the simulated patterns do not fit the experimental ones.

In-plane local vibration

The in-plane local vibration may also reproduce the required splitting (i.e., 46.1 kHz and 81.2 kHz). The following are the simulated patterns based on the in-plane local vibration with different vibration angles. Based on this motion model, the vibration angles (2σ) of 62° and 122° can reproduce the patterns with the distance of 46.1 kHz between two horns, while the vibration angles (2σ) of 45° can reproduce the patterns with the distance of 81.2 kHz between two horns. Figure 6 shows the vibration model and the simulated patterns based on this motion model with the different vibration angles.

Figure 6. The vibration model and the simulated patterns based on this motion model with the different vibration angles.

The characteristics of the pattern lineshapes in Figure 6 are similar to those in Figure 3. We made the similar summation procedure to reproduce the pattern with three sets of horns (i.e., one set of outer horns and two sets of inner horns). But the patterns do not fit the experimental ones (The inner horns in the simulated patterns are too wide to fit those in the experimental patterns).

After many attempts, we realized that the sharp inner horns in the experimental patterns are not very likely to be reproduced via the fast motion models. Based on the pattern lineshapes and the characteristics of the pattern lineshape changes upon temperature, we think that the motion model in the intermediate motional regime

would be the more suitable motion model to describe the motion of imidazolium-d₃. This consideration yields the motion model used in the pattern simulation in the manuscript. But note that, because only the limited motion models were considered, we cannot completely exclude the possibility to reproduce the patterns via a superposition of the signals from the fast and slow motions as that described in the papers ([dx.doi.org/10.1021/jp012391p](https://doi.org/10.1021/jp012391p) and [dx.doi.org/10.1063/1.458354](https://doi.org/10.1063/1.458354)).

Finally, we want to note that the states of the guest molecules in the papers ([dx.doi.org/10.1021/jp012391p](https://doi.org/10.1021/jp012391p) and [dx.doi.org/10.1063/1.458354](https://doi.org/10.1063/1.458354)) are different to the state of the guest molecules in our work. In the paper ([dx.doi.org/10.1021/jp012391p](https://doi.org/10.1021/jp012391p)), the benzene molecules confined in the meso-pores form two different phases in an individual pore, i.e., the surface phase and the inner core phase. The superposition of the two signal components is resultant from the molecular motions in the two phases. Similarly, in the JCP paper by Roessler et al. (doi: 10.1063/1.458354), the guest molecules (hexamethylbenzene and benzene) are trapped in the polymer and form the two phase domains too. These scenarios are completely different from that in our case which is a single-phase crystalline system and, in the crystal, each anionic cage only contains one guest molecule cation. Characterizations indicate that no different phases coexist at one specific temperature.

I was shocked to learn that adding simply a rigid spectral component in the simulation

reduces the preexponential factor from $6.13 \times 10^{18} \text{ s}^{-1}$ to $7.44 \times 10^{12} \text{ s}^{-1}$. That means that values of the activation energy and of the preexponential factor may be strongly affected by systematic errors. It is, therefore, not appropriate to publish so exact numbers which may change strongly depending on the way of how the analysis is done.

Reply:

As to the preexponential factor and activation energy, we re-simulated the patterns with the parameters used in the pattern simulation for the previous version of the manuscript and realized that there was a mistake in the simulation. In the second simulation, this mistake has been corrected. This contributes to the change in the pre-exponential factor, i.e., preexponential factor from wrong $6.13 \times 10^{-18} \text{ s}^{-1}$ to the present $7.44 \times 10^{12} \text{ s}^{-1}$ as the Reviewer questioned. Moreover, in the second simulation, more motion frequency values are obtained from the pattern simulation. This yields the much more reliable activation energy and the pre-exponential factor than those from the first simulation.

We are very sorry for this mistake that arouses the confusion.

I hope that the authors will take this comment into account, so that I will be able to recommend publication of a suitably revised manuscript.

Reply:

We greatly appreciated the Reviewer's valuable comments!

Reviewer #4 (Remarks to the Author):

The manuscript reports the novel H/D effect for the structural transition of host-guest crystals. The author examined the deuterated effects of the dielectric phase transition for molecular motion of Im cation, where the D replacement expanded the host hydrogen-bonding lattice. The essential point of the inverse D/H effect is the unusual host lattice transformation coupled with the motional freedom of cation.

Although such phenomenon should be one of the interesting phase transition mechanism, the motional freedom of Im cation has not been clearly discussed as pointed out by the reviewer. The detail simulation results of NMR measurements for each D isomer should be carefully mentioned, and should be reviewed by specialist of solid state NMR.

Reply:

The discussion of the motional freedom of the Im cation was carefully revised. Additional descriptions and figures were added into the manuscript and the supporting information, such as

"Due to the geometric restriction from the crystal lattice, the energetic favorable motion model of the Im cation is the in-plane motions."

"The pattern simulation reveals that the Im cations likely undergo an in-plane two-site jump motion with a jump angle of 95° (Supplementary Fig. 11)."

In the manuscript, we choose the **1-d₃** as a representative for the NMR measurements.

As to **1-d₂** and **1-d₅**, they are not good candidates because of the existence of the O—D···O hydrogen bonds that would contribute to the lineshapes and make the interpretation much more complicated.

In minor point, the standard deviation should be included in hydrogen-bonding distances.

Reply:

The standard deviation was included in hydrogen-bonding distances in Figure 5.

After the minor revision above, the paper should be progressed in the publication of Nature Comm.

Reply:

Thanks!

REVIEWERS' COMMENTS:

Reviewer #3 (Remarks to the Author):

The authors have considerably improved their paper. The comments I raised have been taken into account. The authors state now that they can not exclude possibility of a superposition of medium/fast rotating and rigid molecules. That is ok. However, for their own sake, the authors should add a sentence that their interpretation is based on the assumption of a crystalline single phase system, which can not exhibit a coexistence between different phases at one specific temperature (see author reply). If a given method observes only a single phase, it might be that this method is not precise enough to detect two very similar coexisting phases. There is one example in the literature where NMR saw two phases whereas x-ray analysis only gave a single phase (<http://dx.doi.org/10.1021/ja002688l>).

I recommend the manuscript for publication after minor revisions, I do not need to see it again.

Reviewer #3 (Remarks to the Author):

The authors have considerably improved their paper. The comments I raised have been taken into account. The authors state now that they cannot exclude possibility of a superposition of medium/fast rotating and rigid molecules. That is ok. However, for their own sake, the authors should add a sentence that their interpretation is based on the assumption of a crystalline single phase system, which cannot exhibit a coexistence between different phases at one specific temperature (see author reply). If a given method observes only a single phase, it might be that this method is not precise enough to detect two very similar coexisting phases. There is one example in the literature where NMR saw two phases whereas x-ray analysis only gave a single phase (<http://dx.doi.org/10.1021/ja002688l>). I recommend the manuscript for publication after minor revisions, I do not need to see it again.

Reply:

Thanks for this comment!

We added sentences just above the Discussion section in the revised manuscript:

"However, note that the interpretation of the NMR results is based on the assumption that only a single crystalline phase exists in the system. In principle, the NMR results can also be interpreted by a superposition of the signals from different crystalline phases that have the same X-ray diffraction patterns but different ^2H NMR pattern features⁵⁰. Based on the above X-ray and NMR observations, we cannot completely exclude this possibility."